# Neural Network Modelling of Temperature and Salinity in the Venice Lagoon

Fabio Bozzeda [1,2], Marco Sigovini [3] and Piero Lionello [3,*]

1 Department of Biological and Environmental Sciences and Technologies, University of Salento, 73100 Lecce, Italy; fabio.bozzeda@unisalento.it
2 National Biodiversity Future Center, 16126 Palermo, Italy
3 National Research Council, Institute of Marine Sciences, Arsenale Tesa 104, Castello 2737/F, 30122 Venice, Italy; marco.sigovini@ve.ismar.cnr.it
* Correspondence: piero.lionello@unisalento.it

**Abstract**

This study applies an artificial neural network (ANN) to simulate monthly temperature and salinity variations at three stations in the Venice lagoon, which have been selected to represent different regimes (marine, riverine and intermediate) in terms of relevance of local processes and exchanges with the open sea. Four key predictors are shown to play a major role: mean offshore sea level, 2 m air temperature, precipitation for the lagoon water temperature, integrated with offshore sea surface salinity for the lagoon water salinity. The development of the ANN is based on only 4 years of observations, taken irregularly over time with an approximately monthly frequency. Despite this, the ANN achieves an accurate reproduction of both variables with large $R^2$ and reasonably small, normalized root-mean-square errors at all stations, except for the salinity at the marine station, where the model presents a spurious variability, which is absent in observations. Sensitivity analysis shows that the 2 m air temperature is the dominant predictor for water temperature while sea-level and sea surface salinity are the principal predictor of salinity fluctuations, with precipitation exerting a relevant role mainly at the riverine station. The ANN has been used for a set of synthetic climate change analyses considering 1.5, 2 and 3 °C global warming levels with respect to preindustrial levels. An overall warming of lagoon water with maximum increase in summer is expected (up to 6 °C in the 3 °C global warming level), resulting in an amplification of the annual cycle amplitude. The expected increases in salinity have a strong gradient across the lagoon, are largest at the riverine station, and (analogously to the changes in temperature) amplify the salinity annual cycle amplitude.

**Keywords:** Artificial Neural Network; Venice Lagoon; climate change; temperature; salinity

## 1. Introduction

Coastal lagoons, such as the Venice Lagoon, represent highly dynamic and ecologically significant environments that provide essential ecosystem services, including habitat provisioning, nutrient cycling, and coastal protection [1–3]. The Venice Lagoon, the subject of this study, is a paradigmatic example of such a vulnerable system. It is characterized by a shallow, microtidal environment with a complex morphology of salt marshes, tidal flats, and deep channels. For decades, it has been subjected to significant anthropogenic pressures, including industrial pollution, nutrient loading leading to eutrophication, and morphological alterations from dredging and coastal defenses. These pressures have made its ecosystem particularly fragile. In this context, surface water temperature and salinity act

as master variables, directly influencing species distribution, metabolic rates, and the overall biogeochemical functioning of the lagoon. Therefore, accurately modeling their present and future dynamics is fundamental for understanding the resilience of this unique ecosystem. These transitional systems act as biodiversity hotspots, supporting a wide range of species, including commercially and ecologically valuable fish and invertebrates [4]. Rising global temperatures are leading to higher water temperatures, altering species distribution and metabolic rates, while changes in evaporation and precipitation patterns influence salinity gradients and nutrient dynamics [5,6]. Sea-level rise poses a direct threat to these low-lying coastal systems by altering tidal regimes and increasing the risk of coastal erosion and habitat submersion [7]. Additionally, marine heatwaves are becoming more frequent and intense [8] leading to physiological stress in aquatic organisms and altering trophic interactions [9–12]. Projections suggest that these climate-induced pressures will intensify in the coming decades, potentially surpassing the impacts of traditional non-climatic drivers and leading to irreversible ecological shifts in coastal lagoons worldwide [13,14]. This study considers the Venice Lagoon, an environment that is increasingly threatened by the combined effects of climate change and anthropogenic pressures, leading to profound alterations in their physical, chemical, and biological characteristics [15]. Understanding and assessing the future impact of climate change on coastal lagoons is crucial for developing adaptive management strategies to enhance the resilience of the lagoons [16].

### 1.1. Rationale for a Data-Driven Modeling Approach

The selection of predictors to model the lagoon's surface water temperature and salinity is grounded in the fundamental physical processes governing these systems. This study uses a set of atmospheric and marine variables as predictors to simulate the lagoon's internal response. It is crucial to distinguish between 2 m air temperature (T2), used as an input predictor, and surface water temperature, which is the output variable the model aims to predict. This study proposes an artificial neural network (ANN) for simulating salinity (S) and temperature (T) variations in the Venice Lagoon. The analysis is based on data collected from three monitoring stations (Figure 1) within the lagoon, chosen to be representative of the internal variability of this complex and dynamic coastal system [17]. The locations of these stations were identified by a based on local hydrodynamic conditions, considering freshwater inflows and water exchanges with the sea [18]. The selected stations represent key transitional areas within the lagoon, where the influence of freshwater inputs, tidal exchange, and meteorological variability combine to create distinct hydrographic patterns [19] (Figure 1). The selection of predictors to model lagoon temperature and salinity is grounded in the fundamental physical processes governing the heat and salt budgets of these coastal systems. Atmospheric forcings are represented by 2 m air temperature and air humidity, which create the thermodynamic gradients for sensible and latent heat fluxes, respectively, along with the 10 m wind components that modulate the rate of these air-water exchanges [20]. Wind, in particular, is a critical driver of vertical mixing and circulation in shallow lagoons [21] and directly influences evaporation rates [22], thereby impacting both temperature and salinity. The hydrological balance is addressed through precipitation, a direct input of freshwater that causes dilution [23], and sea level, which dictates the volume and timing of water exchange with the adjacent sea [24]. This exchange is the primary mechanism for the advection of marine properties into the lagoon. Finally, for the salinity model, sea surface salinity is included as it critically defines the salt concentration of the marine water exchanged with the lagoon [25]. The predictor set for the temperature model does not include this variable. Collectively, this predictor set provides a physically comprehensive framework to capture the dominant atmospheric, terrestrial, and marine drivers of hydrographic variability within the lagoon. The investigation of

parameters such as water salinity and temperature in these systems has traditionally relied on process-based dynamic models. While physically grounded, the practical application of these models is often constrained by their substantial computational expense, which arises from the need for very fine spatial resolutions to resolve complex hydrodynamic features. This computational burden is exemplified by the implementation of advanced numerical models like the System of Hydrodynamic Finite Element Modules (SHYFEM). As detailed by Micaletto et al. (2022) [17], running such a model requires significant High-Performance Computing (HPC) resources. For instance, in performance tests for a regional configuration (the SANIFS system), a simulation covering just a 7-day period required the use of the 'Zeus' supercomputer, a parallel machine equipped with dual 18-core Intel Xeon Gold CPUs per node. The benchmarks reported by the authors involved utilizing up to 288 cores, corresponding to eight full computational nodes, for this short-term simulation. This level of resource allocation underscores the prohibitive cost of conducting the long-term, multi-decadal, and large-ensemble projections that are essential for a robust assessment of climate change impacts. In light of these computational constraints, data-driven methods present a viable alternative.

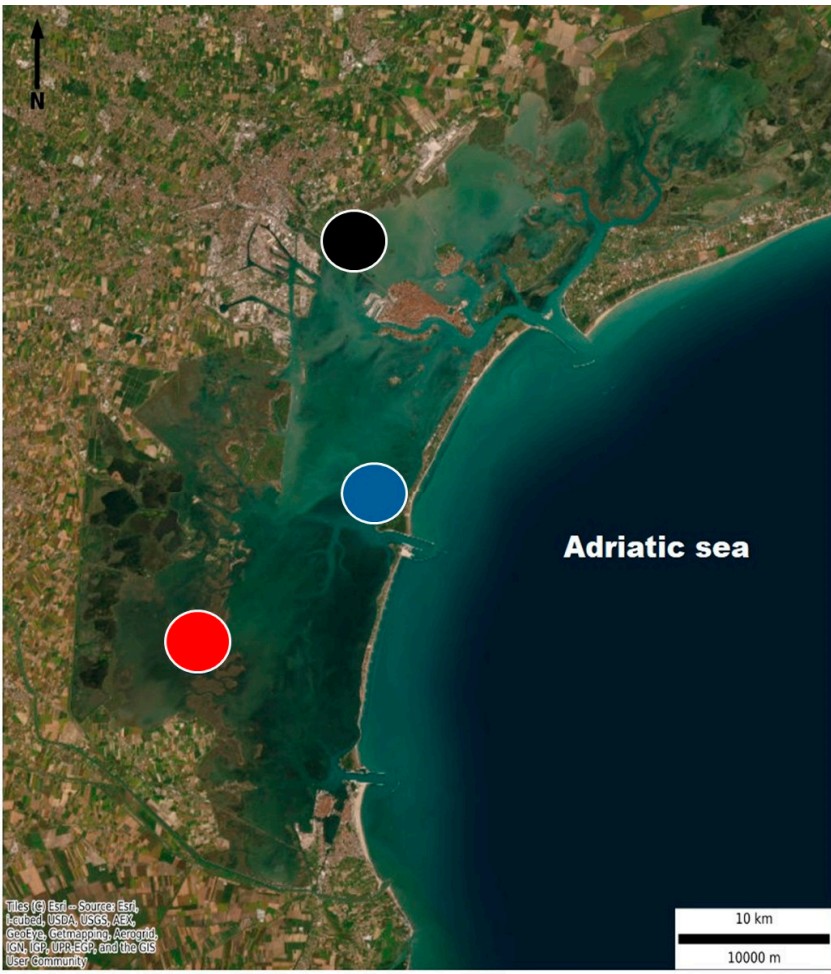

**Figure 1.** Map of the Venice Lagoon showing the locations of the three monitoring stations: riverine (black), marine (blue), and intermediate (red).

### 1.2. Convolutional Neural Networks as a Novel Tool for Coastal Hydrodynamics

This study explores an approach centered on an artificial neural network (ANN). ANNs have an established history of application in hydrological and climate contexts, having been used for tasks such as the bias correction and statistical downscaling of

daily precipitation (e.g., Chen et al., 2010 [18]). Within the broader family of ANNs, this work leverages a Convolutional Neural Network (CNN), an architecture that has shown considerable promise in climate science due to its inherent capacity for processing spatially organized data. In contrast to traditional ANNs that operate on tabular data, CNNs are structured to effectively process gridded datasets, making them well-suited for analyzing climate model outputs and geographical information, as benchmarked in frameworks like WeatherBench [26]. Their utility stems from the ability to automatically learn and exploit spatial hierarchies in data—distinguishing between large-scale drivers and local-scale features—and to recognize patterns with a degree of locational invariance.

The application of CNNs is informed by a growing body of research in regional climate modeling, where these methods have been shown to offer competitive performance. They have been successfully employed for spatial downscaling, learning to translate coarse-resolution General Circulation Model (GCM) outputs into high-resolution regional projections. For example, the DeepSD model [27] demonstrated the efficacy of CNNs for generating high-resolution precipitation fields, and a similar approach was validated by Baño-Medina et al., (2020) for downscaling ERA5 reanalysis data. Furthermore, CNNs are used to create computationally inexpensive "emulators" of physics-based Regional Climate Models (RCMs), enabling the rapid generation of ensemble projections that would otherwise be infeasible [28]. Advanced architectures, such as U-Nets, or techniques like transfer learning have further enhanced the ability of CNNs to preserve complex spatial structures and generalize across different domains [29,30]. However, despite their proven utility in these large-scale atmospheric and terrestrial applications, the application of CNNs to model the fine-scale, tidally driven hydrodynamics of key parameters like salinity and temperature within a complex coastal lagoon represents a novel application of these techniques. The proven success of artificial neural networks (ANNs) in modeling large-scale atmospheric and terrestrial systems provides the motivation for exploring their potential in more computationally demanding domains. Fine-scale coastal hydrodynamics, particularly in tidally dominated environments like the Venice Lagoon, represents one such frontier. Here, the modeling objective shifts from capturing broad spatial patterns to resolving highly localized phenomena, such as salinity fronts and thermal stratification within complex channel-shoal systems.

### 1.3. Model Architecture and Study Outline

The model developed in this work is a CNN architecture designed to capture the spatiotemporal dynamics of the Venice Lagoon. The core of the model consists of two hidden convolutional layers, each containing 30 neurons, which facilitates the hierarchical extraction of spatial features from the input variables. A known challenge for deep learning models is their dependency on large and diverse datasets to generalize effectively. This is particularly relevant in environmental science, where long-term, high-resolution monitoring data can be scarce. To address this limitation, the model's training process is augmented with a Reinforcement Learning-like sequential optimization mechanism. This strategy was implemented to enhance the model's ability to learn robustly from a limited database by allowing it to iteratively refine its predictive policy based on sequential feedback. The resulting framework (Figure 2) provides an adaptable tool for potential use in ongoing environmental monitoring and data-driven decision support in vulnerable coastal ecosystems. Beside this introduction, this paper contains a "Data and Methods" section that describes the artificial neural network (ANN) model, the datasets used, and their sources and investigates multicollinearity among potential predictors. Section 3, "Results", describes the identification of a robust and minimal set of predictors, a sensitivity analysis and the ANN validation. Section 3 includes also an application using a synthetic dataset,

based on existing studies on climate change [31] to demonstrate the effectiveness of the ANN model for climate analysis. The final section, "Discussion and Conclusions", discusses the outcomes of the study, limitations and the potential applications of the ANN model.

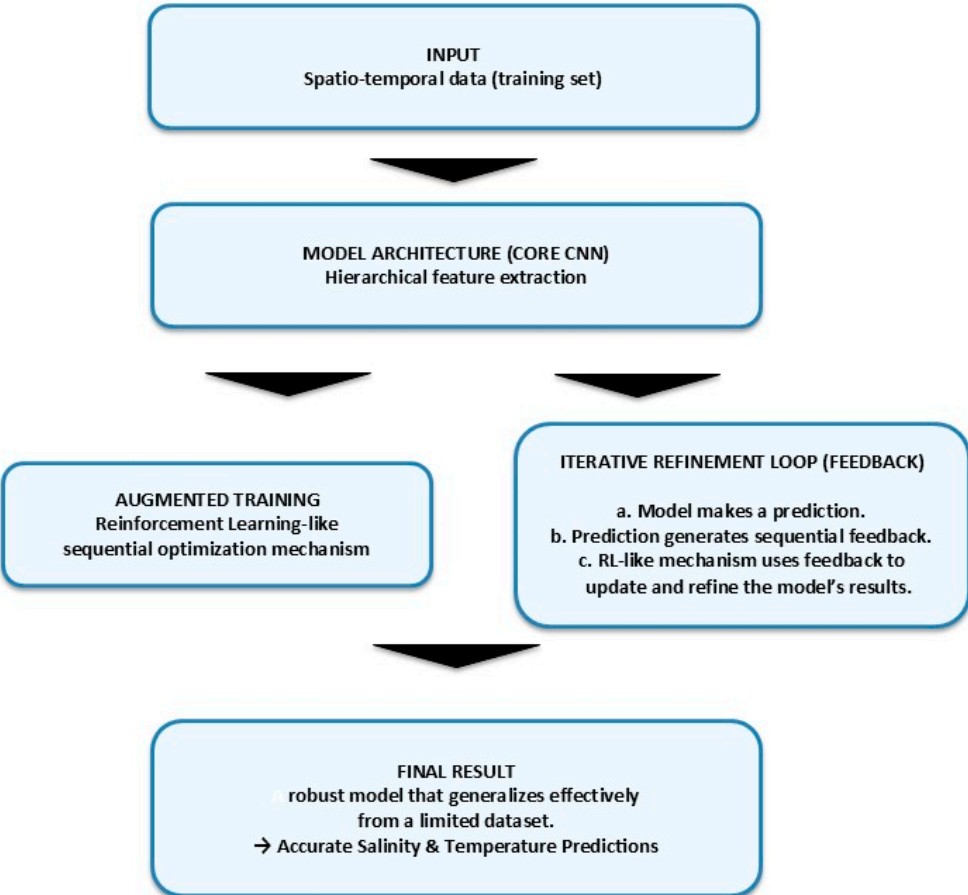

**Figure 2.** Flowchart of the proposed model architecture and training process. The diagram illustrates the workflow of the Convolutional Neural Network (CNN) model developed to simulate the hydrodynamics of the Venice Lagoon. The model takes as input spatio-temporal data, which are processed through a core architecture of two hidden convolutional layers for hierarchical spatial feature extraction. To address the challenge of a limited dataset, the training process is improved with data augmentation tecniques. This introduces an iterative feedback loop where the model's predictions are used to sequentially refine its predictive capability, enhancing its ability to generalize effectively and produce robust predictions of key parameters such as salinity and temperature.

This study aims to address these challenges by pursuing three primary objectives. First, to develop and validate a novel Convolutional Neural Network (CNN) framework (Figure 2) capable of accurately simulating fine-scale salinity and temperature dynamics in a complex coastal system using a limited set of physically interpretable predictors. Second, to leverage this validated model to conduct a sensitivity analysis, identifying the dominant environmental drivers that govern the hydrography in different regions of the Venice Lagoon. Finally, to apply the model to generate quantitative projections of future hydrographic conditions under established climate change scenarios, thereby demonstrating its utility as a computationally efficient tool for climate impact assessment. Beside this introduction, this paper is structured as follows. Section 2, "Data and Methods", describes the datasets, the model architecture, and the experimental design. Section 3, "Results", presents the model's performance, the sensitivity analysis, and the climate change projections without interpretation. Section 4, "Discussion", interprets these findings, contextualizes them within existing literature, and addresses the study's limita-

tions. Finally, Section 5, "Conclusions", summarizes the key outcomes and suggests future research directions.

## 2. Data and Methods

Surface temperature and salinity observations at the three selected stations were produced within the framework of the MELa Project, carried out by Magistrato alle Acque di Venezia (Venice Water Authority) through the concessionaire Consorzio Venezia Nuova. The project monitored the lagoon on a monthly basis producing a database of the main physical, chemical water parameters and key ecological indicators of biodiversity. In this study we use the temperature and salinity data at the three stations (riverine, marine, and intermediate, Figure 1), which are available for the period from September 2001 to December 2004. The energy and water surface mass fluxes, integrated by the exchanges with the open sea across the lagoon inlets, represent the phenomenological basis for the computation of the temperature and salinity inside the lagoon [32]. In this study we consider a set of meteorological variables affecting the surface evaporation flux (air temperature at 2 m level, T2, wind speed zonal and meridional components, Uw, Vw, specific humidity at 2 m level, q2) and precipitation (P). To account for the exchanges with the open sea we consider sea level (SL), sea surface temperature and salinity (SST and SSS, respectively). For all variables the monthly mean values are used, greatly facilitating in future applications downloading and storing data of climate models and reanalyzes. Values of $T_2$, $U_w$, $V_w$, $q_2$ and P have been retrieved from the European Union ERA5 Service [33]. SST, SSS and SL are downloaded from the Copernicus Climate Change Service, +Climate Data Store, (2021). SST and SSS are provided by ORAS5 global ocean reanalysis monthly data from 1958 to present. Copernicus Climate Change Service (C3S) Climate Data Store (CDS). DOI: https://doi.org/10.24381/cds.67e8eeb7 accessed on 1 July 2025. SL values are derived from time series produced using the Deltares Global Tide and Surge Model (GTSM), version 3.0—a hydrodynamic model that simulates water levels at 10 min intervals. All data have been used for computing monthly mean values at the spatial resolution of 0.25° to ensure spatial consistency across variables. At the monthly scale, $T_2$ and SST have a substantially identical variability (their correlation is 0.989, Table 1) and using both of them could very likely increase unrealistically the variance without any advantage. Therefore, SST has not been used as input in the ANN.

**Table 1.** Table showing the correlation coefficients between pairs of predictors.

| Variables | SSS | SL | SST | T2 | P | $U_w$ | $V_v$ | q2 |
|---|---|---|---|---|---|---|---|---|
| SSS | | −0.041 | −0.693 | −0.735 | −0.037 | −0.498 | −0.041 | 0.336 |
| SL | | 1 | 0.391 | 0.325 | 0.413 | 0.017 | −0.063 | 0.181 |
| SST | | | 1 | 0.989 | 0.197 | 0.423 | 0.060 | −0.153 |
| T2 | | | | 1 | 0.170 | 0.497 | 0.075 | −0.196 |
| P | | | | | 1 | 0.057 | 0.159 | 0.561 |
| $U_w$ | | | | | | 1 | 0.577 | −0.053 |
| $V_w$ | | | | | | | 1 | 0.367 |
| q2 | | | | | | | | 1 |

### 2.1. Model Architecture and Training Protocol

The model developed in this study is an artificial neural network (ANN) based on a one-dimensional Convolutional Neural Network (CNN) architecture. CNNs are a class of deep learning models particularly effective at identifying local patterns and hierarchical features within structured data [33]. While conventionally applied to two-dimensional image analysis, their architecture is readily adapted to one-dimensional sequence data,

such as time series, where convolutional filters can learn to recognize temporally localized patterns (e.g., diurnal or seasonal effects).

The model is built upon a one-dimensional Convolutional Neural Network (CNN) architecture, chosen for its proven effectiveness in identifying localized patterns and hierarchical features within time-series data. This makes it particularly suitable for capturing how short-term meteorological events influence the lagoon's hydrodynamics. The core architecture consists of two hidden convolutional layers designed to extract relevant temporal features from the input predictors. The output from these layers is then processed through a dense layer to produce the final continuous prediction of either surface water temperature or salinity. Key components, such as the Rectified Linear Unit (ReLU) activation function and the Adam optimizer, were selected based on standard best practices for computational efficiency and robust model training. A key challenge in environmental modeling is often the limited availability of extensive training data. To address this, a non-standard training protocol was implemented to enhance model robustness and long-term simulation accuracy. Instead of minimizing the error at each individual time step, the model was trained using a sequential optimization strategy. This approach is conceptually related to reinforcement learning principles, where the objective is to learn a policy that minimizes the cumulative error metric over an entire prediction horizon. The model's weights were iteratively updated to reduce the mean squared error calculated across a full forecast window, thereby encouraging the network to learn dependencies that maintain stability over longer time scales, rather than solely optimizing for single-step-ahead accuracy.

The model was implemented in Python (v3.8) using the Keras high-level API with the TensorFlow (v2.x) backend [34]. The Adam optimizer was employed for the weight update process due to its adaptive learning rate capabilities [35].

### 2.2. Experimental Design and Climate Change Assessment

A convolutional neural network was developed with two hidden layers, each composed of 30 neurons (Figure 3), with ReLU activation functions applied to each neuron [36]. In order to construct the ANN, the observed time series have been split into a training period (17 values for each variable and station from September 2001 to December 2002) and a test period (24 values from January 2003 to December 2004). The accuracy of the ANN has been assessed using the Normalized root mean square error (NRMSE), where the normalization factor is the standard deviation of the observations [37]. The sensitivity of the ANN to each input variable has been assessed by calculating the associated out-of-bag NRMSE, that is the NRMSE of the ANN when the time series of the single considered variable is resampled using a bootstrapping technique. Large values of the out-of-bag NRMSE denote variables explaining a large fraction of the variability of the observed time series.

Three global warming levels (GWLs) have been considered, 1.5, 2, and 3 °C, with respect to pre-industrial levels. A set of synthetic climate change simulations have been produced. The time series of the ANN inputs for the period 2001–2004 have been perturbed as a function of the GWL consistently with the results of [37]. $T_2$ was increased using a seasonal factor obtained multiplying the GWL by 0.5 in winter, 1.0 in spring and autumn, and 1.5 in summer. Offshore sea level was raised by 25 cm, 50 cm, and 75 cm for the 1.5, 2, and 3 °C increases, respectively. Precipitation was altered by applying seasonal variation proportional to the GWL: an increase of 4 mm/°C in winter, no change in spring and autumn, and a decrease of 12 mm/°C in summer. Additionally, offshore marine salinity has been increased by 0.15 PSU/°C proportionally to the global warming level.

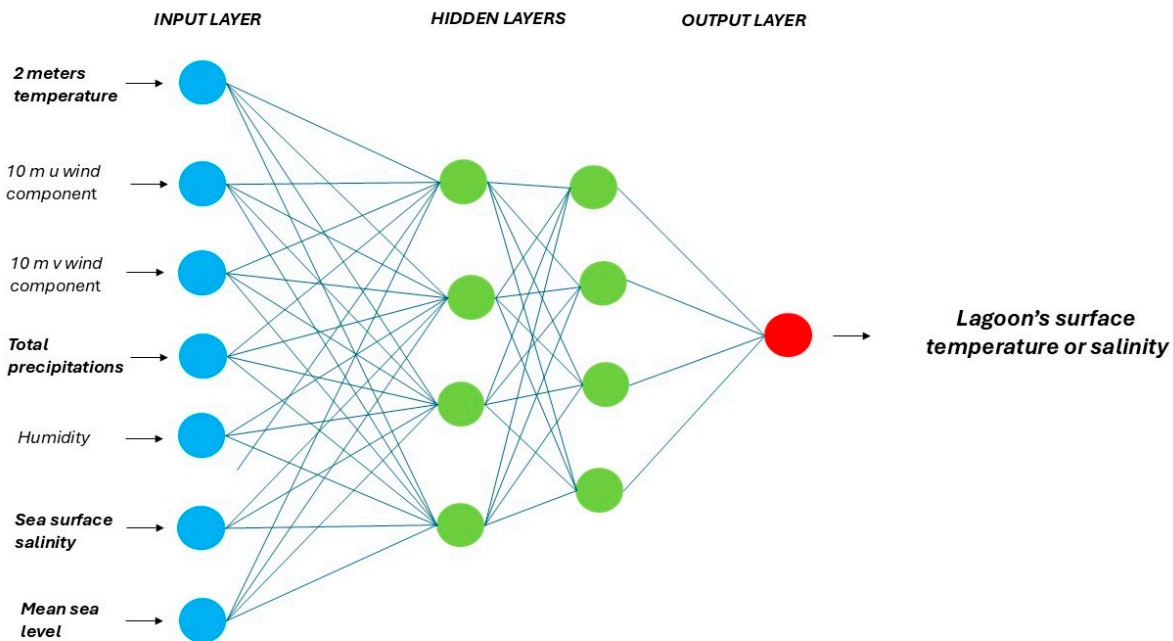

**Figure 3.** Architecture of the convolutional neural network (ANN) used for modeling salinity and temperature variations. The network consists of two hidden layers, each with 30 neurons, employing sigmoid activation functions. The inputs of the ANN are the predictors, which consists of meteorological and oceanographic variables. The predictors that have been selected for the climate change simulations are highlighted in bold: 2 m air temperature, total precipitation, and sea level for the temperature ANN, integrated with the sea surface salinity for the salinity ANN.

## 3. Results

The observed time series of temperature and salinity at the three monitoring stations for the period 2001–2004 are shown in Figure 4. Temperature variations exhibit a consistent seasonal cycle across the three stations, with similar magnitudes and phasing. In contrast, salinity shows significant spatial heterogeneity. The marine station is characterized by high and relatively stable salinity values, whereas the riverine station shows lower, more variable salinity, indicative of brackish conditions. The intermediate station displays values that are between the other two. Inter-monthly salinity variations are notably smaller at the marine station compared to the riverine and intermediate stations. The initial analysis focused on identifying a minimal yet robust set of predictors. A model trained with a full set of six predictors for temperature and seven for salinity achieved high accuracy, with Normalized Root Mean Square Error (NRMSE) values in the test period only marginally higher than in the training period (Figure 5). A sensitivity analysis was then performed to quantify the importance of each predictor (Figure 6, top row). This analysis revealed that a subset of predictors—namely 2 m air temperature (T2), precipitation (P), and sea level (SL) for temperature, plus sea surface salinity (SSS) for salinity—accounted for the majority of the explained variance. Consequently, the models were retrained using only this reduced set of predictors. The results show a very marginal decrease in accuracy (Figure 5), confirming the suitability of the minimal predictor set. The sensitivity analysis for the reduced models (Figure 6, bottom row) confirms that all selected variables have a comparable and significant influence on the predictions. The performance of the final models with the reduced predictor set is further detailed by examining the distribution of errors. The errors for both temperature and salinity show no systematic dependence on the annual cycle, despite the large seasonal variability in both predictors and outputs (Figures 7 and 8). However, the model's accuracy for salinity at the marine station is lower than at the other stations. As shown in Figure 4 (top-right panel) and

Figure 8 (middle panel), the model introduces a spurious variability at the marine station that is not present in the observations, leading to a higher NRMSE (Figure 5). The sensitivity analysis provides quantitative insights into the differing dynamics across the lagoon. At the marine station, which is close to a lagoon inlet, model predictions are most sensitive to SL and SSS. Conversely, at the riverine station, salinity is primarily determined by T2 and P. Finally, the validated ANN was used to project the impacts of climate change under three Global Warming Levels (GWLs) of 1.5 °C, 2.0 °C, and 3.0 °C (Figure 9). The results project a general warming across all stations and throughout the year, with a maximum in summer where the temperature increase is approximately double that of winter. This leads to an amplification of the annual temperature cycle, with summer maximums exceeding 30 °C in the 3.0 °C GWL scenario. Salinity is projected to increase at all stations, with the largest rise occurring in summer at the riverine station (over 4 psu for the 3.0 °C GWL), Figure 10. Similar to temperature, this results in an amplified annual salinity cycle.

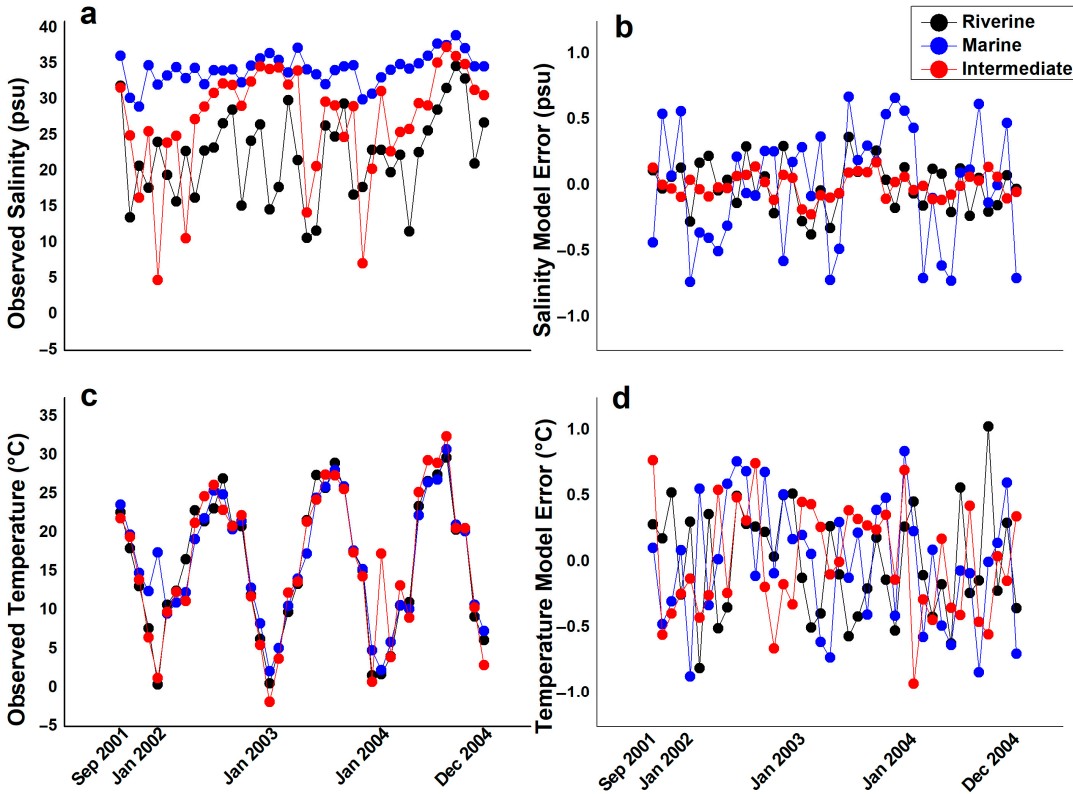

**Figure 4.** Observed time series of salinity (psu, (**a**,**b**)) and temperature (°C, (**c**,**d**)) at the riverine, marine, and intermediate stations from September 2001 to December 2004. The second column shows the corresponding model errors.

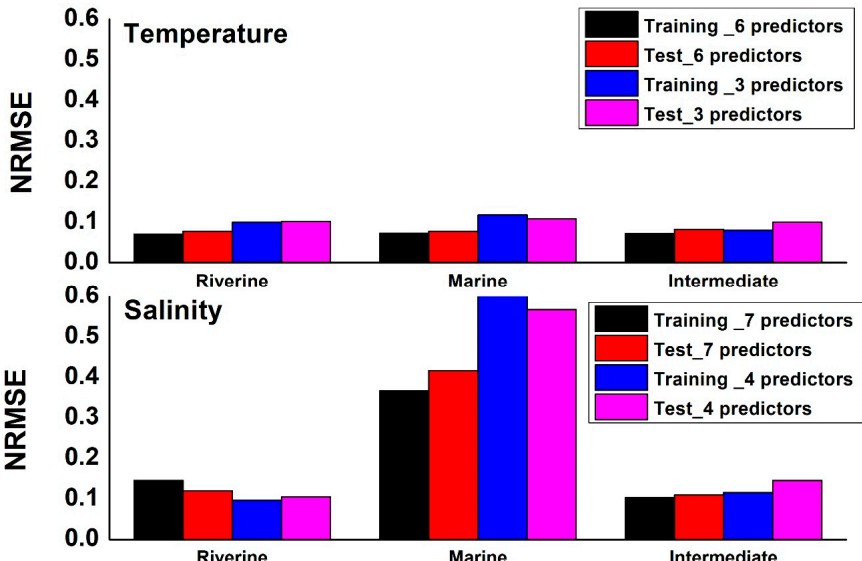

**Figure 5.** Normalized Root Mean Square Error (NRMSE) values for the training and test periods using different numbers of predictors. For temperature simulations, 3 predictors were used, T2, P, SL, to which Uw, Vw, q2 were added for the simulations with 6 predictors. The same variables integrated with SSS have been used for the simulation of salinity with seven and 4 predictors.

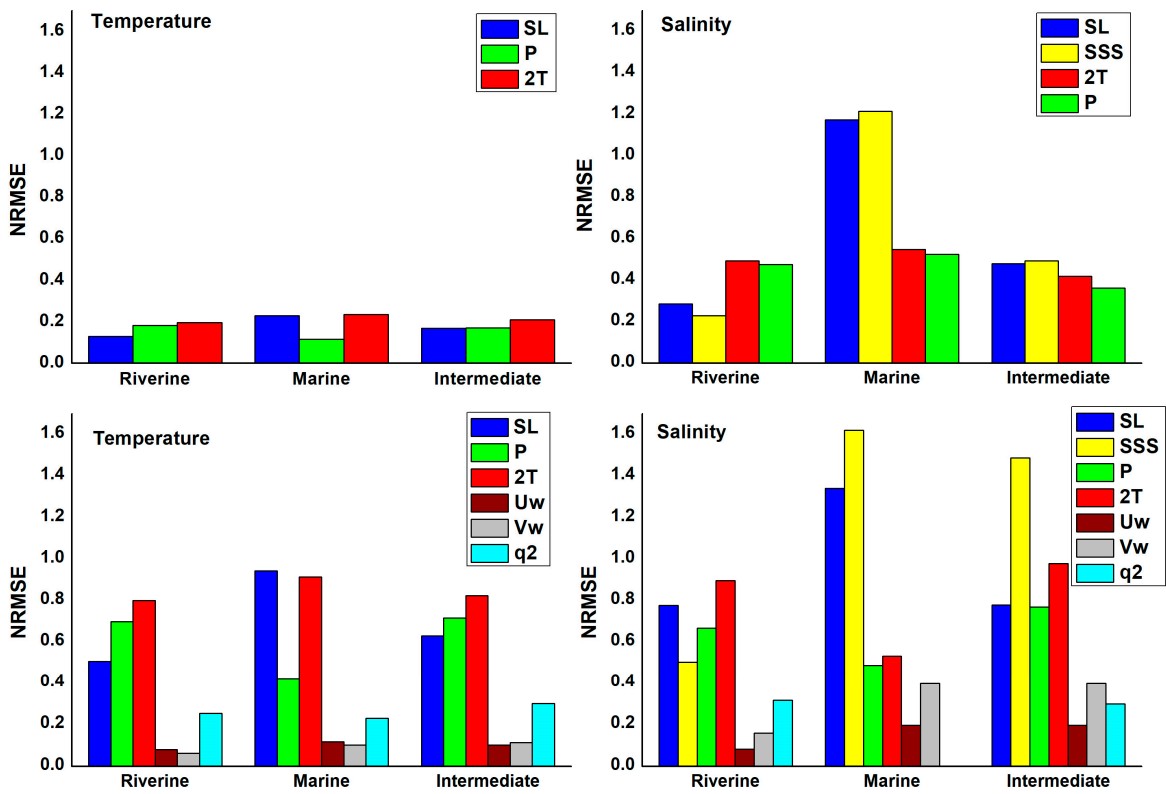

**Figure 6.** Sensitivity analysis results showing the out-of-bag NRMSE. Large values denote large sensitivity. The **top** row shows sensitivity using 3 predictors (for temperature, **left**) and 4 predictors (for salinity, **right**). The **bottom** row shows sensitivity with 6 (for temperature, **left**) and 7 predictors (for salinity, **right**).

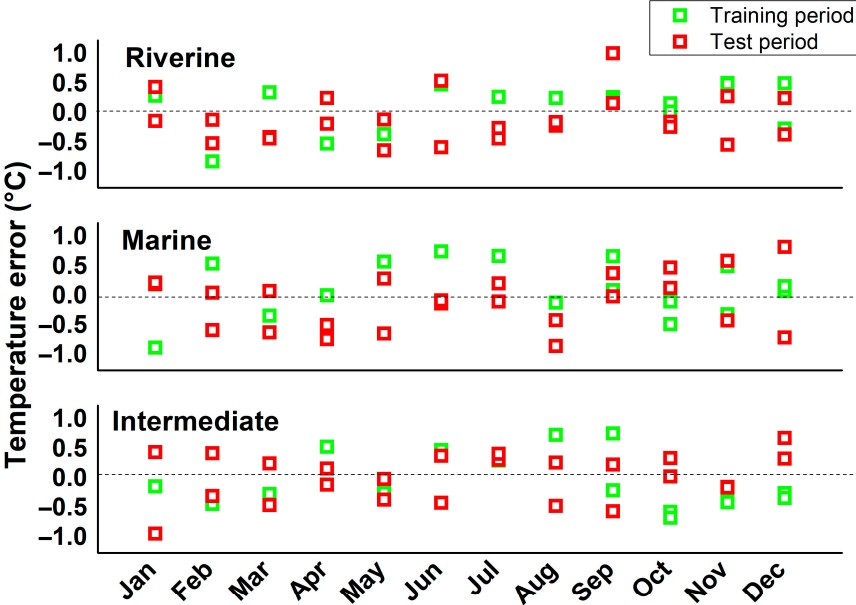

**Figure 7.** Temperature errors (simulated minus observed temperature) at the riverine, marine, and intermediate stations during the training phase (green) and test phase (red) as a function of months.

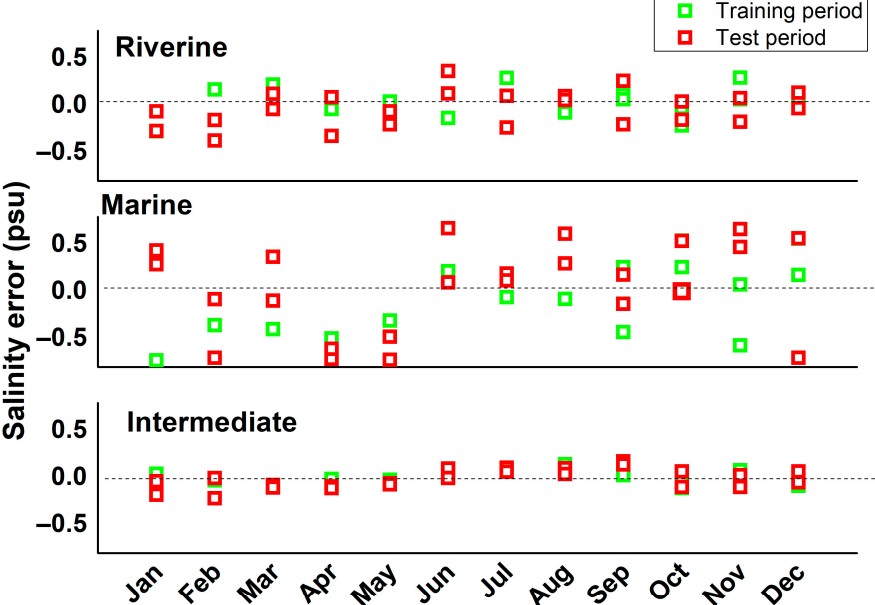

**Figure 8.** Salinity errors (simulated minus observed Salinity) at the riverine, marine, and intermediate stations during the training phase (green) and test phase (red) as a function of months.

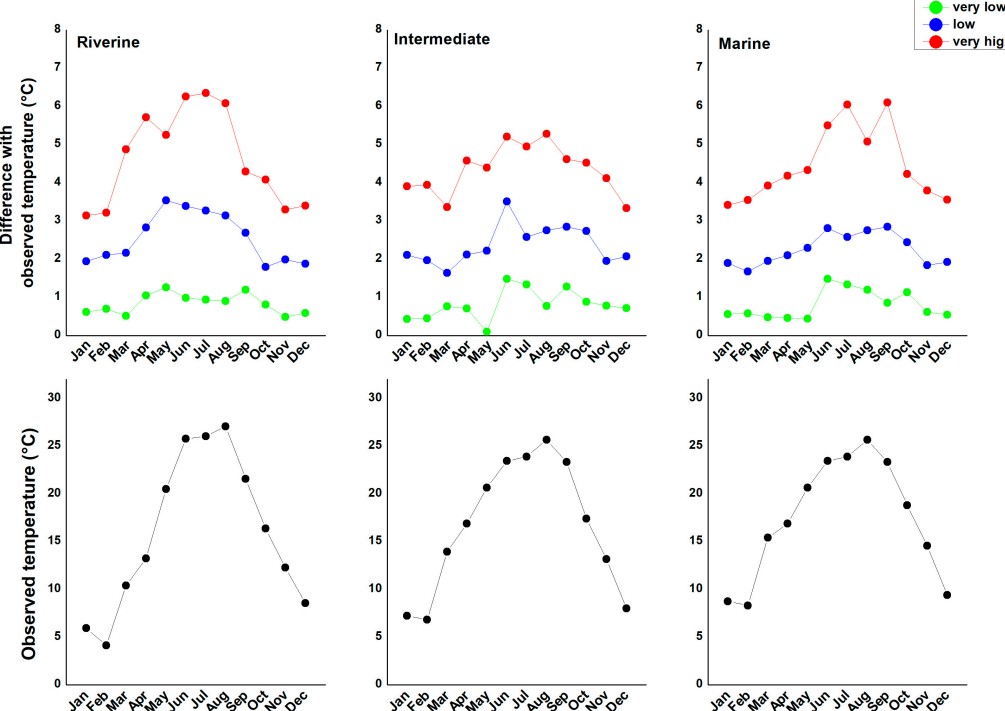

**Figure 9.** Changes in temperature with respect to the observed period (2001–2004) at the riverine (**left column**), marine (**central column**) and intermediate (**right column**) stations under the three global warming levels of 1.5 °C, 2.0 °C, and 3.0 °C.

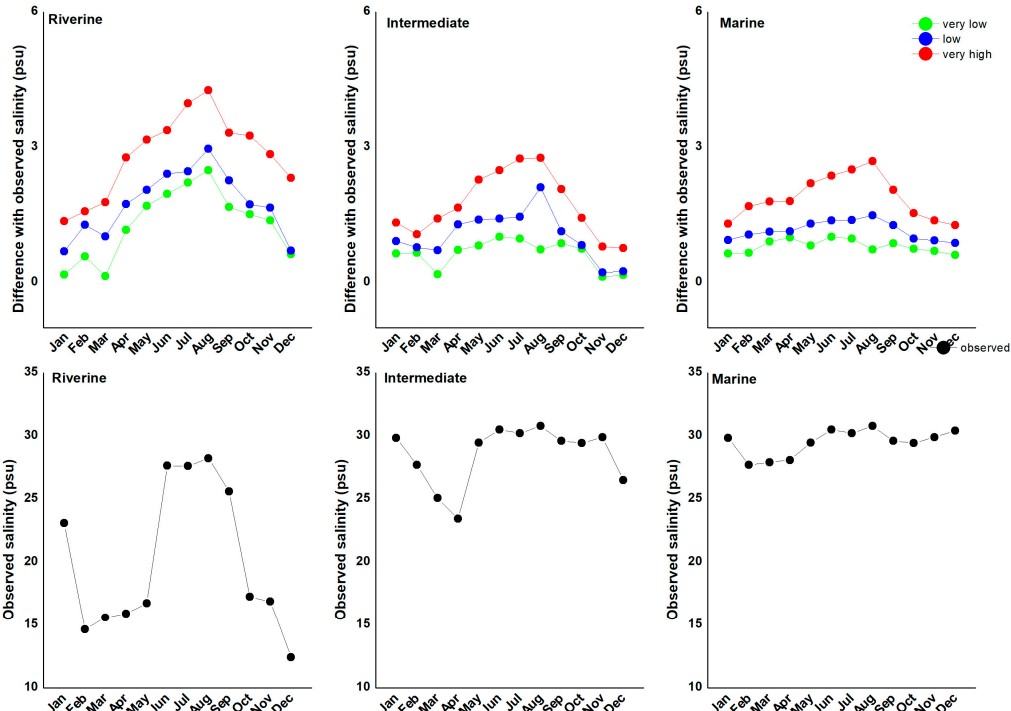

**Figure 10.** Changes salinity with respect to the observed period (2001–2004) at the riverine (**left column**), marine (**central column**) and intermediate (**right column**) stations under the three global warming levels of 1.5 °C, 2.0 °C, and 3.0 °C.

## 4. Discussion

*4.1. Model Performance and Projections of Climate-Driven Impacts*

The results demonstrate that the developed artificial neural network can effectively simulate the complex hydrographic dynamics of the Venice Lagoon. The high performance metrics ($R^2 > 0.85$ for salinity, $R^2 > 0.96$ for temperature) confirm the model's ability to capture the non-linear relationships between a minimal set of environmental drivers and the lagoon's response. This success, achieved with a limited dataset, highlights the power of the CNN architecture and the sequential optimization training protocol. The sensitivity analysis not only allowed for the identification of a robust and minimal set of predictors but also provided physically plausible insights into the lagoon's functioning. The model learned that the marine station's hydrography is dominated by oceanic forcing (sea level and offshore salinity), which aligns with its proximity to a lagoon inlet. In contrast, the riverine station was found to be driven primarily by atmospheric and terrestrial inputs (air temperature and precipitation), consistent with its location near a freshwater source and far from the sea inlets. This alignment between the data-driven model's learned relationships and established hydrodynamic principles suggests the model is capturing true physical processes, not just spurious correlations. The model's lower performance for salinity at the marine station warrants discussion. Observations show this area has low natural variability, acting as a boundary condition influenced by the Adriatic Sea. The model's tendency to introduce spurious variability suggests it may be overly sensitive to small fluctuations in the input sea level and offshore salinity data, which are themselves products of reanalysis models and contain uncertainties.

Sea level and offshore sea surface salinity are by far the most important predictors for the marine station, which is directly impacted by the exchanges with the open sea because of its proximity to a lagoon inlet. The 2 m air temperature and precipitations are the main predictors at the riverine station, which is far from the lagoon inlets, experiences a limited direct water exchange with the open sea and it is close to the mouth of a river with a small basin adjacent to the lagoon–land border. At the intermediate station all four predictors exert a role of comparable importance. The same different dynamics do not emerge clearly from the sensitivity analysis of the temperature simulations, though the role exerted by sea level is larger at the marine station than at the others. To assess potential future changes in the lagoon's hydrography under evolving climatic conditions, the ANN has been forced to use environmental drivers representing future climate conditions. Specifically, the models were used to estimate changes in lagoon salinity and temperature under the expected changes of 2 m air temperature, precipitation, sea level and offshore sea surface salinity. These variables were adjusted according to simplified rates of change established in the scientific literature [37]. By integrating these climate-sensitive predictors, the study provides a forward-looking perspective on how the Venice Lagoon's hydrographic regime may evolve in response to ongoing global climate change. Under the three global warming scenarios (1.5 °C, 2.0 °C, 3.0 °C with respect to pre-industrial), our analysis suggests an overall increase in temperature and salinity, substantially changing the properties of the water masses of the lagoon in the 3 °C GWL. In this high warming scenario increases are over 6 °C temperature and 4psu for salinity in summer at the riverine station, which is the most affected by climate change. The increases in both temperature and salinity have a clear seasonality, being largest in summer with the consequence of substantially amplifying the amplitude of the seasonal cycle. Such large changes in the water masses characteristics may particularly affect benthic communities adapted to spatial niches [38] and that may have limited tolerance to high salinities and temperatures. Increased temperatures and salinities could exacerbate physiological stress on stenohaline species, promote eutrophication through altered stratification, and shift trophic interactions [38,39]. These results

align with global observations of marine heatwaves driving mass mortalities and shifts in species distributions [40], but extend them by quantifying localized lagoon responses. Furthermore, warmer conditions may intensify stratification, reduce oxygen solubility, and elevate eutrophication risks, necessitating proactive management to mitigate harmful algal blooms. The ANN framework thus provides a valuable scenario-based tool for anticipating ecological thresholds and informing adaptive management

### 4.2. Study Limitations and Methodological Context

Despite the promising results, several limitations warrant consideration. However, the ANN approach relies on the availability and quality of input data. In our case only 4 years of observations are available and the sampling of the phenomenology is consequently limited. It cannot be excluded that important links between drivers and responses that would be important in a climate change context are completely missed in the observations used for the training of the ANN. Further the monthly temporal resolution of the used dataset may mask sub monthly events (e.g., storm surges, pulsed river floods) that affect lagoon water properties. However, the ANN has demonstrated a high predictive accuracy, successfully reconstructing hydrographic conditions during the test period and capturing the non-linear interactions that characterize the Venice Lagoon's response to external drivers. The climate change analysis is not directly based on an ensemble of model simulations, but it is based on a simplified reproduction of the ensemble mean resulting from a CMIP5 global ensemble [36]. Considering the underlying non-linear dynamics of the lagoon, this could not represent the outcome of the application of the ANN using the predictors produced by a full ensemble of climate simulations inherent in climate projections. Considering the capability of this approach to depict the actual future of the lagoon temperature and salinity, the ANN does not account for non-climatic anthropogenic pressures, such as land reclamation, navigation dredging, and the effect of closures of the presently operative defense system (MoSE). Therefore, we are aware that further investigations are mandatory, before using these results for prioritize monitoring efforts, designing targeted interventions and assessing the impacts of thermal and salinity extremes. However, this study underscores the potential of data-driven models to complement traditional hydrodynamic simulations, providing a flexible, computationally efficient approach for coastal ecosystem management in a changing climate. The quantitative results indicate that the developed Convolutional Neural Network (CNN) model is capable of modeling the non-linear dynamics of salinity and temperature within the Venice Lagoon. The model achieved performance metrics ($R^2 > 0.85$ for salinity; NRMSE < 0.3 for temperature) that are consistent with the range of values reported in other machine learning applications to coastal hydrodynamics, though direct comparisons are qualified by differences in model architecture and environmental settings [41,42]. The sensitivity analysis suggests that the model's performance for salinity is related to the high predictive importance assigned to offshore sea surface salinity as an input variable.

This study represents, to our knowledge, one of the first applications of a CNN architecture for modeling both salinity and temperature dynamics in a complex lagoonal environment. The necessity for this research is justified by the current state of the field. CNNs are at the forefront of ongoing research due to their advantages in automatically extracting hierarchical and spatio-temporal features from data [43], and they have been increasingly used for applications in climate projections and downscaling. While a growing body of literature uses ANNs for describing salinity and temperature in sub-regional water bodies, very few studies have adopted a CNN approach for this specific task. Previous research on salinity has utilized multilayer feedforward networks [44], Bayesian neural networks [45,46], and hybrid process-driven ANNs [47]. Similarly, temperature modeling

has leveraged deep neural networks, sometimes with transfer learning [48–50]. Within the Venice Lagoon itself, neural networks have been successfully applied to other variables, such as eutrophication [44] and tidal level prediction [51], further demonstrating the potential of these methods in this specific environment.

### 4.3. Methodological Rationale, Interpretability, and Future Directions

The justification for this data-driven approach is further strengthened when contrasted with the computational costs of dynamic models. Finite element models like SHYFEM are a state-of-the-art solution for physically based simulations in the Venice Lagoon. However, their application for long-term climate studies is severely constrained by computational expense. For example, a recent climate change impact study using SHYFEM was limited to only five pairs of 13-month numerical experiments due to these intensive requirements [52–54]. Our CNN framework, once trained, can generate equivalent multi-decadal scenarios in a fraction of the time, enabling the large-ensemble projections necessary for a robust assessment of uncertainty.

The model also appears to have learned physically plausible relationships, as suggested by the sensitivity analysis. The analysis identified distinct sets of drivers for different regions of the lagoon. For the marine-influenced station, sea level and offshore salinity were determined to be the most influential predictors, which is consistent with the known dominance of oceanic forcing. At the riverine-influenced station, 2 m air temperature and precipitation were identified as the most important variables, which aligns with the expected influence of the terrestrial hydrological cycle. This correspondence with established phenomenological drivers suggests that the model is capturing underlying physical processes rather than solely spurious correlations.

The study has several limitations that should be noted. A principal constraint is the four-year dataset with a monthly temporal resolution. This dataset may not fully capture the spectrum of inter-annual variability and may obscure the effects of high-frequency events, such as storm surges or pulsed river floods. Consequently, the model's robustness for predicting conditions significantly outside the distribution of the training data, as might occur under future climate change scenarios, remains unverified. This is a recognized challenge in applying machine learning to Earth system science, where models may not generalize to novel physical states [55].

Furthermore, the interpretability of the neural network model presents a challenge. While the sensitivity analysis provides a high-level overview of variable importance, a more granular attribution of predictor contributions is difficult to extract directly from the model's internal structure [56]. Methodologies from the field of eXplainable AI (XAI), such as SHAP (SHapley Additive exPlanations), provide model-agnostic frameworks to quantify the contribution of each predictor to individual predictions [57]. The application of such techniques is therefore a recommended direction for future research.

## 5. Conclusions

This study successfully developed and applied a novel Convolutional Neural Network (CNN) framework to simulate and project fine-scale salinity and temperature dynamics within the Venice Lagoon. Our work demonstrates that a data-driven approach, augmented with a specialized training protocol, can serve as a computationally efficient and effective tool for assessing climate change impacts in complex coastal ecosystems, even with limited observational data. The primary findings of this research confirm the framework's success on three key fronts. First, the model achieved a high predictive accuracy, successfully reproducing historical salinity ($R^2 > 0.85$) and temperature ($R^2 > 0.96$) variations using a minimal set of physically meaningful predictors. Crucially, its performance extended

beyond statistical accuracy; the sensitivity analysis revealed that the model learned physically plausible dynamics, identifying that oceanic forcing dominates near sea inlets while terrestrial and atmospheric factors are paramount near river mouths. Finally, leveraging the validated framework as a predictive tool, the study projected significant future impacts under a 3.0 °C global warming scenario, with the lagoon expected to experience substantial summer warming (over 6 °C) and increased salinity (over 4 psu in the riverine station), leading to a significant amplification of the seasonal hydrographic cycle. While this study underscores the great potential of data-driven models, future work should focus on integrating higher-frequency data [58] to capture episodic events and incorporating non-climatic drivers, such as the operational regime of the MOSE flood barriers [59,60]. Nonetheless, the framework presented here provides a robust and adaptable tool for environmental monitoring and data-driven decision support in the management of vulnerable coastal systems worldwide.

**Author Contributions:** Conceptualization, F.B. and P.L.; methodology, F.B.; software, F.B.; validation, F.B., M.S. and P.L.; formal analysis, F.B.; investigation, F.B. and P.L.; resources, F.B., M.S. and P.L.; data curation, F.B. and M.S.; writing—original draft preparation, F.B. and P.L.; writing—review and editing, M.S. and P.L.; visualization, F.B.; supervision, P.L.; project administration, P.L.; funding acquisition, P.L. and M.S. All authors have read and agreed to the published version of the manuscript.

**Funding:** This research received no external funding.

**Acknowledgments:** Salinity and temperature data were provided by: Ministero delle Infrastrutture e della Mobilità Sostenibili—Provveditorato Interregionale alle OO. PP. del Veneto—Trentino Alto Adige—Friuli Venezia Giulia già Magistrato alle Acque di Venezia—tramite il concessionario Consorzio Venezia Nuova (Ministry of Infrastructure and Sustainable Mobility—Interregional Authority for Public Works of Veneto, Trentino-Alto Adige and Friuli Venezia Giulia, formerly the Venice Water Authority—through the concessionaire Consorzio Venezia Nuova). Funding was provided by NBFC—National Biodiversity Future Center, funded by European Union—NextGenerationEU, Project code CN_00000033, CUP F87G22000290.

**Conflicts of Interest:** The authors declare no conflicts of interest.

## Abbreviations

List of abbreviations present in the text.

| | |
|---|---|
| ANN | Artificial Neural Network |
| CNN | Convolutional Neural Network |
| GCM | General Circulation Model |
| GWL | Global Warming Level |
| NRMSE | Normalized Root Mean Square Error |
| P | Precipitation |
| RCM | Regional Climate Model |
| ReLU | Rectified Linear Unit |
| SL | Sea Level |
| SSS | Sea Surface Salinity |
| SST | Sea Surface Temperature |
| T2 | Air Temperature at 2 m |
| q2 | Specific Humidity at 2 m |
| Uw, Vw | Zonal and Meridional Wind Components |

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
