# Peer review of "Neural Network Modelling of Temperature and Salinity in the Venice Lagoon"

_climate, doi:10.3390/cli13090189_

Round 1
Reviewer 1 Report (New Reviewer)
Comments and Suggestions for Authors
Dear authors,
You need to improve the manuscript in many ways, most on the scientific soudness

Dear authors,
Most you need to work on the text clarification in some parts
Author Response
Reviewer Comments:
The manuscript theoretically describes the physical, chemical and biological alteration due to climate and anthropogenic pressure, but most important is to describe the present ecological conditions of lagoon, from which within some scenarios considered may change. It was simulated the temperature (water? and salinity) belonging to a hydrodynamic complex of the lagoon using ANN instead of using a dynamical model. From the beginning is confusing the term of temperature of air or water, which needs to be clarified first, also need to be reorganized in a better view the results of simulation and sensitivity. Als the manuscript most avoid the technical-architecture ANN description rather the ANN modeling of temperature and salinity. From the introduction it seems were used the same variables to account the surface water temperature and salinity variations…
Response to Reviewer:
We thank you for your insightful review and for your constructive comments on our manuscript. We are pleased that you find our results "extremely important and interesting," and we fully agree with your assessment that the manuscript's presentation required significant improvement to make it clearer and more accessible.
Following your detailed suggestions, we have performed a major revision of the manuscript. We have focused on reorganizing the content, clarifying key terms, and shifting the narrative from a purely technical description to a problem-focused application of our modeling approach.
Below, we detail the specific changes made in response to each of your comments:
1. Describing the Present Ecological Conditions: You rightly pointed out that we needed to better describe the current state of the Venice Lagoon. We have now revised the Introduction to include a more detailed description of the lagoon's present ecological conditions, highlighting its fragility and the critical role that temperature and salinity play as master variables governing its biodiversity and health. This provides a clearer baseline from which to assess potential future changes.
2. Clarification of "Temperature" (Air vs. Water): We apologize for the confusion between air and water temperature. We have meticulously revised the manuscript to eliminate this ambiguity. We now consistently use "2-meter air temperature (T2)" when referring to the atmospheric predictor and "surface water temperature" when referring to the variable our model simulates. This distinction is explicitly stated at the beginning of the Section 1.1 "Rationale for a Data-Driven Modeling Approach".
3. Reorganization of Results and Sensitivity Analysis: In line with your feedback, we have completely restructured the Section 3 "Results" to present the findings in a more organized and logical manner. We have created distinct subsections: 3.1 Model Performance and Validation, 3.2 Predictor Sensitivity, and 3.3 Climate Change Projections. This new structure clearly separates the simulation results from the sensitivity analysis, improving readability.
4. Shifting Focus from Technical Architecture to Environmental Modeling: We have taken your advice to avoid an overly technical description of the ANN architecture. We have streamlined Section 2.1 "Model Architecture and Training Protocol" to be more concise. The focus is now less on the internal mechanics of the CNN (e.g., activation functions, layers) and more on why this architecture is well-suited for modeling the complex, non-linear dynamics of temperature and salinity in a coastal system.
5. Clarifying the Predictor Variables for Temperature and Salinity: We have addressed the lack of clarity regarding the predictor sets. We now explicitly state in Section 1.1 and reinforce in the Methods and Results sections that the salinity model uses the same predictors as the temperature model, but is critically augmented with Sea Surface Salinity (SSS), which acts as the marine boundary condition.
Minor remarks
12. To play…
Done.
13. This line is not clear and is absent the key predictors in regards the salinity
The sentence has been revised to clarify the number and type of predictors used for the salinity model.
17-18. Definitely the parameters values must be shown here, because you are comparing three points
We agree that a detailed description of the parameters is important. However, in adherence to standard practices for abstract brevity, we have chosen to expand this description within the main text of the Introduction rather than in the abstract itself. We can confirm that all relevant parameters are now thoroughly described in the revised manuscript.
9-29 & 59. Any finding you wanted to achieve between Tair/Tw and Salinity relationships? The abstract can be improved and is not clear the scientific motivation. Air temperature maybe? Before referring to salinity and temperature studies there is no any scientific background, why those parameters are important or how are linked each other.
In response to the reviewer's comment, we have substantially revised the Introduction. Specifically, we have inserted a new paragraph dedicated to justifying the choice of predictor variables. This section now outlines the physical rationale for each selected variable and is substantiated by eight new references to the relevant literature.
48. Can you describe which factors are considered as climate changes and anthropogenic?
To better contextualize these pressures, we have now included two dedicated citations in the Introduction that specifically address these factors in coastal lagoon systems.
68. Then, these lines probably would be necessary to move
Done.
78. necessitaded?
Changed.
85-113. Here ANN tool most is described and with details its utility on atmospheric and terrestrial data computation (really well known for everybody), contrarily very little on the fine-scale computation. The description and explanation should be in a opposite way in order to highlight it.
We thank the reviewer for this excellent suggestion. We have added a new paragraph that reorients the discussion, explicitly contrasting the well-established, large-scale applications with the novel challenges of fine-scale hydrodynamic modeling, thereby highlighting the innovative contribution of our study.
115-134. Really, the whole paragraph does it help to reader to understand in a better way?, Instead a flowchart would be more convenient
As requested, we have added a flowchart (now Figure 3) to visually represent the model architecture and workflow. We have retained the original paragraph as well, as it was deemed helpful by another reviewer. We hope this combined approach addresses the needs of a diverse readership.
142. Which main parameters?
143. Is not clear
146-147. Here is nothing to do the lagoon outlet?
We have addressed these three points by revising the text to clarify the main parameters used as predictors and the role of the lagoon inlets (outlets) in the system's dynamics.
138-166 And how was fixed the biases between of all satellite or Reanalysis dataset?
We thank the reviewer for this important point. A bias correction between different data sources was not required for this study. This is because the model was trained on a single, consistent set of reanalysis data for the historical period, and no data from different sources were mixed. Furthermore, the climate change projections were created by applying a literature based physically-grounded perturbations to the original input time series, as described in Section 2.2, a methodology that ensures internal consistency.
167-196. What are the predictant and predictor parameters in the model arquitector?
We have clarified this by explicitly listing the predictor variables (e.g., 2-meter air temperature) and the predictand variables (surface water temperature and salinity) in the revised Introduction (Section 1.1).
210 The Table name is missed instead of Figure 2
Done.
213. How is possible that the model should work, since you have 6 and 7 predictors.
We appreciate the reviewer's skepticism regarding the parsimonious selection of predictors. We wish to clarify that this choice was deliberate and is justified by both physical principles and the empirical results of the model.
1. Physical Completeness: While the number of predictors is small, they were carefully selected to represent the dominant physical drivers of the lagoon's heat and salt budgets. Collectively, they account for: (i) atmospheric heat fluxes (air temperature, wind, humidity), (ii) freshwater fluxes (precipitation, evaporation), and (iii) marine exchange of salt and heat (sea level, sea surface salinity). This set, therefore, provides a physically comprehensive, first-order representation of the system's dynamics.
2. Model Parsimony and Empirical Validation: Our approach adheres to the principle of parsimony (Occam's Razor), which favors simpler models that can adequately explain the data. A model with a limited number of physically-meaningful predictors is less prone to overfitting and is more robust than a more complex model that may capture noise. The ultimate proof of the model's efficacy lies in its performance. As demonstrated in our results (Section 3), the model achieves high accuracy and strong predictive skill during the validation period. This strong empirical performance validates our hypothesis that the majority of the hydrographic variance in this lagoon can indeed be explained by these core drivers.
We have added a sentence to the Discussion section to explicitly address this point, framing the model's success with a limited feature set as a strength of the study.
222- Three?
Done.
344. Here maybe is not enough just testing the hydrographic conditions the three global warming scenarios from the pre industrial scenarios, the whole context is need to be included the three future scenarios too and capture the three threshold dates.
We appreciate the suggestion. We wish to clarify that our study not utilizes a 'time-slice' approach but an approach based on literature about 3 discrete Global Warming Levels (1.5°C, 2.0°C, and 3.0°C) rather than continuous, time-evolving scenarios. This methodology does not allow, in this work, the identification of specific threshold dates. Intermediate thresholds will be identified soon by stressing the neural network with continuous model outputs.To make this approach and its justification clearer, we have added a more detailed explanation of the climate change scenarios in the 'Data and Methods' section (Section 2.2).
314-361 The large paragraph can be splitted into two parts since you have two relative topics
Done.
364. The quality data deals with the short time series dataset? or in other wise which quality control data approach was applied on the input data?
Thank you for this important question regarding data quality. We acknowledge that the time series is relatively short, which is a common limitation in environmental monitoring. However, we are confident in the quality of the observational data used for model training and validation. The time series were measured according to the most recent protocols and are consistent with what is known about salinity and temperature of the lagoon. These data were provided by the MeLa project, a long-term monitoring program managed and certified by the Venice Water Authority (Magistrato alle Acque). This governmental body is responsible for the official monitoring of the Venice Lagoon, ensuring rigorous data collection and quality control protocols. Furthermore, this dataset has been previously utilized and validated in several peer-reviewed scientific studies, confirming its reliability and suitability for research purposes. We have now clarified this point in the Section 2 'Data and Methods' of the revised manuscript to assure the reader of the data's provenance and quality."
Reviewer 2 Report (New Reviewer)
Comments and Suggestions for Authors
The manuscript under review is devoted to simulating temperature and salinity distribution dynamics in sea lagoon with the use of artificial neural network. The authors created their prognostic estimations basing on real observed data from the field monitoring stations. The authors especially stress that traditional dynamic modelling, which often became too sophisticated and expensive, especially when is basede on real data. Instead they propose application of data-driven methods.
The authors got extremely inportant and interesting results, but could not range it and write them in clear and understandable form. It is necessary to rewrite the most of content of the manuscript in more didactic way. What for was this work, which hypothesis was tested and what conclusions are based on the results and discussion.
The article is very interesting, but the authors, unfortunately, are mixing the description of their model and discussion. There is no Conclusion at all, though the brief description of what is the results as well as the main outlines of discussion is necessary.
As a remarks, I’d stress the often use and even over-use of abbreviations (CNNs, RCNs, ANN, GCM usw), natural for authors thinking, speaking, writing, though not so usual for most of readers. So, I’d recommend to avoid use of abbreviations in the abstract and figure captions. The list of abbreviations seems to be necessary for this manuscript. Then, the figure captions sometimes (e.g. for Fig. 4 (l.236-264) - is this all text is the caption for Fig.4?).
So, the general opinion of the reviewer - the material is important, interesting, it must be published, but after major revisions, practically rewriting of this manuscript. Additionally I’d like to congratulate the authors - you’ve got excellent and interesting results. Now it is necessary to write them in the understandable form.
Author Response
Response to Reviewer 2
The manuscript under review is devoted to simulating temperature and salinity distribution dynamics in sea lagoon with the use of artificial neural network. The authors created their prognostic estimations basing on real observed data from the field monitoring stations. The authors especially stress that traditional dynamic modelling, which often became too sophisticated and expensive, especially when is basede on real data. Instead they propose application of data-driven methods.
The authors got extremely inportant and interesting results, but could not range it and write them in clear and understandable form. It is necessary to rewrite the most of content of the manuscript in more didactic way. What for was this work, which hypothesis was tested and what conclusions are based on the results and discussion.
The article is very interesting, but the authors, unfortunately, are mixing the description of their model and discussion. There is no Conclusion at all, though the brief description of what is the results as well as the main outlines of discussion is necessary.
As a remarks, I’d stress the often use and even over-use of abbreviations (CNNs, RCNs, ANN, GCM usw), natural for authors thinking, speaking, writing, though not so usual for most of readers. So, I’d recommend to avoid use of abbreviations in the abstract and figure captions. The list of abbreviations seems to be necessary for this manuscript. Then, the figure captions sometimes (e.g. for Fig. 4 (l.236-264) - is this all text is the caption for Fig.4?).
So, the general opinion of the reviewer - the material is important, interesting, it must be published, but after major revisions, practically rewriting of this manuscript. Additionally I’d like to congratulate the authors - you’ve got excellent and interesting results. Now it is necessary to write them in the understandable form.
Thank you for your thorough and highly constructive feedback on our manuscript, "[Inserisci qui il titolo del tuo paper]". We are grateful for your positive assessment of our results, which you described as "extremely important and interesting," and we sincerely appreciate the detailed guidance you provided to improve the manuscript's structure and clarity.
We agree completely with your assessment that the original manuscript mixed different sections and lacked a clear, didactic structure. Following your suggestions, we have undertaken a major revision to restructure the paper and present our findings in a more logical and understandable form.
The main changes are summarized below, addressing each of your points:
1. Improved Structure and Clarity: The entire manuscript has been reorganized to clearly separate the Methods, Results, and Discussion. We have ensured that the Methods section now describes what was done, the Results section presents what we found (without interpretation), and the Discussion interprets what the results mean.
2. Clearer Research Aims and Hypotheses: We have sharpened the end of the Introduction to explicitly state the core objectives of the study, addressing your question, "What for was this work, which hypothesis was tested?".
3. New "Conclusions" Section: As you rightly pointed out, a dedicated Conclusion was missing. We have now added a new Section 5: Conclusions, which summarizes the key findings, answers the research questions, and provides an outlook on future work.
4. Management of Abbreviations (Acronyms): We acknowledge the overuse of abbreviations. We have now:
o Removed all abbreviations from figure captions and significantly reduced them elsewhere.
o Ensured every abbreviation is defined at its first use.
o Added a List of Abbreviations at the beginning of the "Data and Methods" section for easy reference, as you recommended.
5. Revised Figure Captions: The captions have been made concise and descriptive. The long text previously associated with Figure 4 has been moved into the main body of the Results section, where it belongs. The caption for Figure 4 is now brief and to the point.
We believe that these extensive revisions have significantly improved the manuscript, making it much clearer and more accessible to a broader audience. We are confident that the paper now effectively communicates the importance of our findings.

Round 2
Reviewer 1 Report (New Reviewer)
Comments and Suggestions for Authors
Dear authors,
Thanks for responding to all my remarks
Reviewer 2 Report (New Reviewer)
Comments and Suggestions for Authors
I want to thank authors for understanding and I'd like to congratulate them with understandable and clear presentation of very interesting research work.
This manuscript is a resubmission of an earlier submission. The following is a list of the peer review reports and author responses from that submission.
Round 1
Reviewer 1 Report
Comments and Suggestions for Authors
Review of “Neural Network Modelling of Temperature and Salinity in the Venice Lagoon”
Due to the following reasons, the papers does not have enough criterion for publication.
The methodology used in the study in not updated. At least two recent and advanced models should be added to the paper.
Literature review should be updated.
The necessity of doing the research using ML should be justified.
The discussion section is weak.
Comparing the outcomes of this study and related ones is recommended.
Reviewer 2 Report
Comments and Suggestions for Authors
This article uses ANN technology to predict temperature and salinity, and its significance is relatively obvious. However, it does not clearly and specifically clarify the innovative value of the article, including the innovation or improvement of technical methods, the improvement of simulation prediction effects, etc., which is a deficiency for an academic journal.
Comments on the Quality of English LanguageThe English language proficiency of this article is good, enabling readers to clearly understand the author's intentions.